# Using Network Science to Evaluate Landslide Hazards on Big Sur Coast, California, USA

Vrinda D. Desai<sup>1,2</sup>, Alexander L. Handwerger<sup>3,4</sup>, and Karen E. Daniels<sup>1</sup>

Correspondence: Vrinda D. Desai (vddesai@ncsu.edu)

Abstract. Landslides, ranging from slips to catastrophic failures, pose significant challenges for prediction. This study employs a physically inspired framework to assess landslide hazard at a regional scale (Big Sur Coast, California). Our approach integrates techniques from the study of complex systems with multivariate statistical analysis to identify areas prone to landslide hazards. We successfully apply a technique originally developed on the 2017 Mud Creek landslide and refine our statistical metrics to characterize landslide hazard within a larger geographical area. Our method is compared against factors such as landslide location, slope, displacement, precipitation, and InSAR coherence using multivariate statistical analysis. Our network analyses, which incorporates spatiotemporal dynamics, perform better as a monitoring technique than traditional methods. This approach has potential for real-time monitoring and evaluating landslide hazard across multiple sites.

## 1 Introduction

15

With climate change leading to extreme weather conditions, such as heavy precipitation, there is an increased global danger of landslide hazards (Kirschbaum et al., 2020). One of the biggest challenges in real-time landslide hazard assessment is identifying and quantifying the likelihood of landslides within a geographical area (Hungr et al., 2014; Palmer, 2017) due to the inherent variability of hillslopes presenting non-uniform spatiotemporal dynamics (Lacroix et al., 2020; Glastonbury and Fell, 2008).

On the coast of California, landslides are abundant due to mechanically weak rocks, active uplift, and high seasonal precipitation, all of which contribute to general instability throughout the area. Hundreds of landslides have been identified as precipitation-induced by exploring the relationship between rainfall and landslide velocity (Handwerger et al., 2022; Scheingross et al., 2013; Bennett et al., 2016; Handwerger et al., 2019; Young, 2015; Booth et al., 2020; Wills et al., 2005; Jones et al., 2019; California Department of Conservation, 2023). As water infiltrates the ground, the water table rises, leading to an increase in pore-water pressure. This rise in pore-water pressure reduces the effective normal stress (difference between normal stress and pore-water pressure), which in turn decreases the frictional strength of the hillslope. This instability can lead

<sup>&</sup>lt;sup>1</sup>Physics Department, North Carolina State University, Raleigh, NC, USA

<sup>&</sup>lt;sup>2</sup>North Carolina Institute for Climate Studies, Asheville, NC, USA

<sup>&</sup>lt;sup>3</sup>Joint Institute for Regional Earth System Science and Engineering, University of California Los Angeles, Los Angeles, CA, USA

<sup>&</sup>lt;sup>4</sup>Jet Propulsion Laboratory, California Institute of Technology, Pasadena, CA, USA

to rapid mass movement of material (rock, earth, debris) down the hillslope, defined as a *landslide*. Identifying areas that are at immediate risk can help focus resources on the analysis, prevention, and risk reduction of these landslides.

In the winter of 2022-2023 (W22-3), the Big Sur Coast witnessed acceleration of four deep-seated landslides. Three of these four landslides – Dani Creek Slide (Mar 2023), Mill Creek (Jan 2023), and Gilbert's Slide (Mar 2023) (Drabinski and Bertola, 2023b) – were recorded by the California Geological Survey and the U.S. Geological Survey (California Department of Conservation, 2023) but were not catastrophic. The fourth landslide occurred in Jan 2023: Paul's Slide, a deep slow-moving landslide, reactivated where some material from the surface failed and engulfed Highway 1 (Drabinski and Bertola, 2023a).

In this study, we apply network science techniques to identify patterns that evolve over space and time (Kivela et al., 2014; Mucha et al., 2010; Porter and Gleeson, 2016). A network framework gives us the ability to infer the underlying dynamics of a creeping hillslope by capturing the relationships and complexities of spatiotemporal interactions even in the absence of detailed, particle-scale information. Instead, we use larger-scale information about the spatiotemporal heterogeneities as a proxy to provide insights into susceptibility and rheological dynamics. Network techniques have been applied across a broad range of physical, biological, and social contexts (Barrat et al., 2008; Nguyen et al., 2019; Newman, 2010; Bassett et al., 2011; Nguyen et al., 2019), including the study of granular and disordered materials (Papadopoulos et al., 2016; Nabizadeh et al., 2022; Berthier et al., 2019) and landslides (Tordesillas et al., 2018; Mei et al., 2025; Zhou et al., 2022; Wang et al., 2025).

The material beneath a hillslope consists of individual grains; the key lesson learned from the granular studies is that granular failure has a transitional period between stable deformation and catastrophic failure, and this is indicated by a distinct dynamical pattern that has been observed in landslides (Singh and Tordesillas, 2020; Tordesillas et al., 2018; Dai et al., 2020; Tordesillas et al., 2024). Building on the success of network science in characterizing failure within granular systems, we previously developed a method to mathematically describe this type of hillslope through spatiotemporal relationships (Desai et al., 2023). In our approach, the hillslope is described as a set of geospatial points (nodes) connected by lines (edges), where each edge encodes measurable information about the relationship between nodes. In the context of landslides and this study, such information is derived from remote sensing observations. Network science enables us to analyze landslide dynamics beyond examination of overall deformation.

The objective of this paper is to integrate the techniques developed in (Desai et al., 2023) with statistical analyses to classify geographical regions as *stable* or *hazard-prone* based on the likelihood of a landslide.

We first tested these network techniques on Mud Creek, a slow-moving deep-seated landslide on the Big Sur Coast that experienced acceleration on 20 May 2017. In our earlier study, Desai et al. (2023), we used multilayer networks (Papadopoulos et al., 2016; Porter and Gleeson, 2016) and community detection methods (Mucha et al., 2010; Porter et al., 2009) to retrospectively identify Mud Creek's location pre-failure and detect the transition from creeping to catastrophic failure.

In this paper, we extend these methods across a broader section of the Big Sur Coast, shown in Fig. 1[a], using velocity from Interferometric Synthetic Aperture Radar (InSAR) time series and slope from a Digital Elevation Model (DEM). We evaluate the effectiveness of the network-based approach by comparing it to multivariate analysis using physical variables: topography, ground surface deformation, precipitation, and InSAR temporal coherence.

**Figure 1. Regional scale maps of study area.** (a) Digital elevation model of the study area. The 4 recorded landslides by USGS and CA are shown as white dots. The black outlined boxes are the 17 sub-regions we used in this analysis. (b) Topographic slope angle (degrees). (c) Cumulative displacement from Sentinel-1 InSAR from Nov 2015 to Dec 2022. (d) InSAR temporal coherence map. (e) Accumulated precipitation from PRISM from 1 Nov 2015 to 30 Nov 2022.

## 2 Data

We used ground surface deformation, topographic characteristics, and precipitation, which are commonly used in landslide hazard maps and forecasting. We also include InSAR temporal coherence as a variable to measure the reliability of the InSAR measurement quality after time series inversion. The following sections discuss these variables.

# 60 2.1 Study Area

The Big Sur Coast is divided into 17 sub-regions of similar size, roughly 5 km<sup>2</sup>, and shown in Fig. 1[a]. This allows for the application of the technique on more varied terrain containing both stable and unstable hillslopes (see discussion in §5.1), as well as for testing the method as a prototype of how this technique might be used in a monitoring context.

## 2.2 Topography

We used the Copernicus Digital Elevation Model (DEM) (Copernicus, 2021), downsampled to  $40 \times 40 \text{ m}^2$  resolution (matching to InSAR resolution) (Hogenson et al., 2020) on a temporal interval of 6 or 12 days (corresponding to Sentinel-1 passes) and is shown in Fig. 1[a], to calculate the slope of each gridded cell within the study area, as depicted in Fig. 1[b]. We write the elevation field as h(r), where r = (x, y) represents the position in UTM coordinates.

### 2.3 Ground Surface Deformation

We processed the Sentinel-1 InSAR data using the Alaska Satellite Facility's (ASF) On-Demand InSAR processing HyP3 platform (Hogenson et al., 2020). ASF's On-Demand InSAR tool constructs interferograms using the GAMMA software. The InSAR data are inverted to time series using the Miami InSAR Time-series software in Python (MintPy) software (Yunjun et al., 2019). We processed data on descending track 42 using two looks in azimuth and ten looks in range. Spanning from 20 Nov 2015 to 01 Dec 2022, the dataset comprises 279 time slices at a resolution of (40 × 40) m<sup>2</sup> per grid cell, covering the study area of the Big Sur Coast. From the Sentinel-1 data, we utilized the displacement time series and a temporal coherence map in our analysis.

Each InSAR image provides line-of-sight displacement, indicating motion either towards or away from the satellite. Cumulative displacement, calculated as the difference between the last time slice on 01 Dec 2022 and the first time slice on 20 Nov 2015, shows landslide activity over the seven years (as shown in Fig. 1[c]). Additionally, we computed the mean and maximum cumulative displacement values of each sub-region for use in multivariate analysis.

To analyze the motion of landslides for use in the network science techniques, we calculated the velocity for each InSAR image t as  $v(x,y,t)=\frac{\Delta u(x,y)}{\Delta t}$ , where  $\Delta u$  represents the relative displacement between pairs of adjacent snapshots and  $\Delta t$  denotes the time interval between any two consecutive snapshots. Due to the noise introduced by taking derivatives of the InSAR displacement data, we apply a  $3\times 3$  cell Gaussian kernel to smooth the data.

The temporal coherence map (Fig. 1[d]) measures the reliability of the InSAR displacement time series inversion. Pixels with coherence values below 65% were masked from the displacement maps. Low coherence values can occur due to a number of factors, including obscured or significant deformation (Fletcher et al., 2007; Yunjun et al., 2019). The applied mask excludes approximately 1.7% of the total area, with an average exclusion of 4.8% within each sub-region.

## 2.4 Precipitation

The Parameter Elevation Regressions on Independent Slopes Model (PRISM) database provides precipitation data modeled on a  $4 \times 4 \text{ km}^2$  grid cell resolution using station data and climate models (Oregon State University, 2015). We selected PRISM due to its performance in mountainous and coastal areas of the western United States and because there are no station data within the study area (Daly et al., 2008). The precipitation datasets span from 1 Nov 2015 to 30 Nov 2022 with daily maps, and we computed cumulative precipitation by summing over the entire period (as shown in Fig. 1[e]), with the mean precipitation calculated for each sub-region.

### 2.5 Reported or Identified Landslides

The U.S. Geological Survey (Jones et al., 2019) and the California Geological Survey (California Department of Conservation, 2023) have reported and identified landslides, compiling these into the Reported California Landslides Database gathered from various local, state, and federal agencies. Although this database is not exhaustive, the entries encompass shallow landslides, slow-moving landslide activity, rockfalls, and debris flows. The four landslides of W22-3, represented as white points in Fig. 1[a], are located from north to south at Dani Creek Slide, Paul's Slide, Mill Creek, and Gilbert's Slide. Each of these landslides involved seasonally-related mass downslope movements of hillslope material. To provide a rough estimate of the surface area of each landslide, we measured the landslides on Google Earth. Dani Creek Slide estimated to be 3,000 m<sup>2</sup>, Mill Creek 4,000 m<sup>2</sup>, and Gilbert's Slide 6,700 m<sup>2</sup>. In contrast, Paul's Slide, a notably slow-moving landslide that reactivated in Jan 2023, underwent over a year of repairs, is estimated to be 60,000 m<sup>2</sup> (Drabinski and Bertola, 2023a, b, c).

#### 3 Methods

### 3.1 Network Science

To classify sub-regions as *hazard-prone*, we used multilayer networks (Kivela et al., 2014; Porter and Gleeson, 2016; Papadopoulos et al., 2016; Bassett et al., 2011) to couple the temporally-resolved kinematics (velocity) with the spatially-resolved susceptibility (slope) and community detection (Porter et al., 2009; Mucha et al., 2010; Fazelpour et al., 2023) to cluster regions that are moving at relatively higher speeds and/or are on steeper slopes. A visual representation of this method is shown in Figure 3 of Desai et al. (2023). Our methods were developed in Desai et al. (2023), and are briefly summarized here (code is available at Desai (2022)).

In the multilayer network, each layer corresponds to an InSAR image that overlies the network topology; the network topology consists of a static collection of nodes and edges for each sub-region. Each node represents a patch of area, determined using Poisson sampling, and edges connect the nodes via Delaunay triangulation. Velocity and slope maps from InSAR and DEM, respectively, are incorporated into the network as edge weights that change for each layer in the multilayer network.

In prior work, we calculated the average velocity and slope of any two connected nodes and set that as the edge weight. We successfully identified clusters (patches of area) exhibiting similar hillslope movements that are distinct from surrounding areas via a community detection algorithm that uses modularity optimization (Desai et al., 2023). GenLouvain (Mucha et al., 2010; Jeub et al., 2011-19), a modularity optimization algorithm, divides nodes into communities by identifying where the edge weights are stronger within the community than one would expect at random. This algorithm outputs a matrix consisting of the community ID for each node and time pair; the community ID is a unique numeric identifier to for each of the different communities across the entire spatiotemporal map (multilayer network). Within this spatiotemporal community map, we identified areas of persistent communities; these communities are an indicator of areas that are strongly connected and have higher-than-average edge weights – velocity and slope – than the rest of the sub-region.

## 3.2 Community Persistence

135

To quantify steady communities, we previously developed a measure known as community persistence (Desai et al., 2023). This measures the persistence of the assignment of nodes to communities over time defined as

130 
$$\Pi_t = \frac{1}{N} \sum_{c} \frac{c_{t-1} \cap c_t}{n_{c,t}},$$
 (1)

where N is the total number of nodes in the network,  $n_{c,t}$  is the number of nodes in community c at time t, and  $c_{t-1} \cap c_t$  is the number of nodes present in community c at both t, t-1.

An increase in community persistence within a sub-region indicates that a group of nodes is consistently more strongly connected to each other (measured via edge weight) than to the rest of the network, corresponding to localized motion. Conversely, low persistence occurs when no distinct, consistently clustered motion is observed, such as during dry seasons.

As done in our prior work (Desai et al., 2023), we compare relative changes in  $\Pi$  across the 17 sub-regions using the Z-score at time t

$$Z_t = \frac{\Pi_t - \bar{\Pi}}{\sigma(\Pi)} \tag{2}$$

where  $\Pi$  is the mean persistence over the entire time period, establishing a baseline of the system, and  $\sigma(\Pi)$  is its standard deviation. When  $Z_t < 0$ , community persistence is below average, typically indicating dry conditions with little landslide activity. In the prior analysis of Mud Creek, we observed that surrounding communities exhibited a drop in persistence, while those within the Mud Creek zone increased in persistence. This led to a statistically significant increase in  $Z_t$ , peaking at Z=2.5, followed by a minor decline prior to failure. The value at failure remained statistically significant. To extend this analysis to the 17 sub-regions, we identify the final rising segment before the decline and extract the *peak* Z value, which is the maximum Z-score in the identified segment. We use PEA to quantify differences between sub-regions to better classify the regions as stable (peak Z < 2.5) or hazard-prone (peak  $Z \ge 2.5$ ).

### 3.3 Multivariate Correlation

We characterized active landslides using data on precipitation, deformation, topography, and InSAR coherence. These indicators were used as benchmarks to evaluate the performance of the community detection results. Specifically, we constructed a

correlation matrix to assess the relationship between the peak Z-score and the geophysical indicators. For each of the 17 subregions, we computed the following seven variables: number of recorded landslides; mean slope; mean precipitation; mean displacement; maximum displacement; mean InSAR coherence; mean community persistence  $\bar{\Pi}$ ; and peak Z-score. These metrics were used to quantify and compare the classification of hazard-prone regions with observed landslides.

### 4 Results

We used a multilayer network combining slope and velocity data spanning from Nov 2015 to Dec 2022 (just before the observed landslides in Jan 2023 and Mar 2023) to determine the success of evaluating landslide hazard via network science techniques. We visualize the evolution of  $Z_t$  for each sub-region in Fig. 2. The size of the data points is proportional to  $Z_t$ , with larger circles indicating more persistent communities (higher Z-scores). Only points with  $Z_t > 0$  are plotted, as negative Z-scores do not signify landslide hazard. The colors in Fig. 2[a-b] correspond to the 17 sub-regions shown in Fig. 1[a].

Over the period shown, we observe cyclical changes in  $Z_t$  corresponding to the wet and dry seasons. Following wet seasons,  $Z_t$  is stronger and after dry seasons,  $Z_t$  is weaker. Particularly, we observe high  $Z_t$  values in fall of 2022; this is because the study area experienced higher than average precipitation that year. The final 100 days of the period (Fig. 2[b]) display differences in Z-scores across the sub-regions, aiding in the identification of potentially stable and hazard-prone areas. A clear distinction is made between sub-regions with an increasing Z-scores (outlined in black) and those exhibiting relatively stable Z-scores during those last 100 days.

As previously discussed, a continuous increase in Z-score, like that observed in Mud Creek, indicates high likelihood of a landslide. To quantify the trends in Fig. 2[a-b], we identify the peak Z of the final continuously increasing segment and plot the values in Fig. 2[c], where sub-regions Z 

Figure 2. Z-score for community persistence. (a) The Z-score for community persistence  $Z_{\Pi t}$  for a multilayer network containing information from Nov 2015 to Dec 2022. Each color represents a sub-region. The size of the points corresponds to  $Z_t$ , where the thicker the point, the higher the Z-score. (b) Time from Aug 2022 to Dec 2022 is shown (inset from [a]), where regions with a continuous increasing segment are outlined in black. (c) Spatial plot of peak Z-score, with regions colored from blue to red. The landslides that occurred in W22-3 are initialed in [b] and shown as white dots in [c], with the corresponding images taken by CalTrans (Drabinski and Bertola, 2023a, b, c).

**Figure 3. Correlation of Landslide Variables.** Correlation for mean precipitation, mean displacement, maximum displacement, mean slope, mean coherence, mean community persistence, peak Z-score of community persistence, and number of landslides. Variables are scaled from minimum to maximum, with darker colors indicating higher values.

recorded landslides (0.63), peak Z-score (0.56), and negatively with InSAR coherence (-0.56). Moreover, precipitation has a strong positive correlation with recorded landslides (0.63) and peak Z-score (0.67).

## 5 Discussion

Through the integration of network science techniques, remote sensing data, and multivariate analysis, this study identifies several key insights into the detection and characterization of landslide-prone regions. The results highlight the promise of the peak *Z*-score as a method for early detection and classification of hazardous sub-regions. This finding builds on recent work examining kinematic patterns in failure zones using clustering and network-based techniques – such as complex networks (Tordesillas et al., 2018), persistent homology (Mei et al., 2025), feature engineering (Zhou et al., 2022), and neural networks (Wang et al., 2025) – to identify landslide precursors and offered alternatives to threshold-based modeling. The present work extends those applications by distinguishing between stable and unstable slopes over a large spatial domain, representing a step toward transitioning from static susceptibility mapping to dynamic, spatiotemporal prediction at regional scales. Such advancements have been enabled by the increasing availability and resolution of InSAR data, which provide a global framework for using satellite radar observations to detect precursory deformation and achieve near–real-time early warnings (Dai et al., 2020; Tordesillas et al., 2024; Wang et al., 2025).

Because our analysis was conducted within a single geographic region and based on a limited number of documented landslides, the outcomes should be viewed as a demonstration of potential rather than a universally generalizable result. Further validation across different terrains, climates, and geologic settings will be essential to establish the broader applicability of this approach as a monitoring tool. Future work might also draw on forecasting frameworks such as Cascini et al. (2022), which emphasize the temporal evolution of displacement trends and provide a pathway toward continuous predictive monitoring.

### 5.1 Spatial Scale

Our analysis was conducted at a subregional scale of 5 km², larger than typical landslide areas of a few 0.1 km², in order to incorporate a mix of stable and unstable terrain within each subregion. Such variation is critical for the modularity optimization algorithm to identify clustered anomalous behavior (Zhou et al., 2022; Mei et al., 2025; Das and Tordesillas, 2019; Singh and Tordesillas, 2020). The presence of stable hillslopes provides a baseline against which unstable slopes transitioning to catastrophic failure can be detected as outliers, given that nearby hillslopes experience similar weather conditions. Reducing the size of the subregion would limit the amount of stable terrain, undermining the algorithm's ability to detect relative deviations. Conversely, increasing the spatial extent would significantly raise computational demands without improving performance or sensitivity. This trade-off between spatial scale and computational efficiency is especially important when identifying early signals of slope reorganization that precede catastrophic failure. A potential direction for future work is to assess whether applying this method at smaller spatial scales could improve the model's performance. While such an approach might enhance its utility for real-time landslide monitoring, it would also reduce terrain heterogeneity, potentially diminishing the algorithm's capacity to distinguish anomalous behavior. If this method were to be operationalized as a monitoring tool, a detailed investigation of how terrain uniformity and subregion size affect performance would be essential.

## 215 5.2 Comparison of Network-Based Metrics and Geophysical Indicators

The correlation analysis highlights important relationships between geophysical variables, network-based metrics, and precipitation-induced landslide activity. Notably, the peak Z-score shows a strong positive correlation with recorded landslides, maximum displacement, and precipitation. This suggests that peak Z is an effective proxy for capturing patterns indicative of an increased potential for landslides, particularly in response to seasonal hydrologic forcing.

Conversely, the negative correlation between both mean and max displacement and InSAR coherence supports previous findings that landslides moving rapidly often evade detection by InSAR (Yunjun et al., 2019). Although slope exhibits a weak relationship with  $\bar{\Pi}$ , it shows a strong correlation with mean displacement. This indicates that the multilayer network is not biased towards steep slopes but rather amplifies steeper hillslopes undergoing deformation. Collectively, these findings demonstrate that network-based metrics – community persistence and peak Z – not only align with traditional indicators of landslide hazard but additionally captures evolving structural patterns in hillslope dynamics which improve classification of hillslopes as hazardous.

Furthermore, several distinct features emerge when comparing the spatial variability along the eight coastline graphs in Fig. 3. Both mean and maximum displacement prominently highlight the sub-region containing Paul's Slide and Dani Creek

Slide. However, notable differences exist throughout the rest of the study area. While measuring mean displacement tends to smooth over localized or sudden deformation and instead emphasizes areas with multiple slow-moving landslides, maximum displacement highlights acute deformation signaling higher instability. Peak Z similarly emphasizes Paul's Slide as a high-risk region – validated by a major slip off Highway 1 at Paul's Slide in Jan 2023, occurring just a month after the analysis period.

Despite both maximum displacement and peak Z-score highlighting Paul's Slide, differences between the two variables is evident in other sub-regions. Some sub-regions with the highest peak Z-score do not exhibit the greatest displacement, and sub-regions with similar displacement magnitudes show a wide range of peak Z-scores. These discrepancies suggest that community detection is capturing additional mesoscale dynamics, such as localized shifts in slope behavior or evolving stability, that are not solely influenced by landslide deformation or slope susceptibility.

While the volume removed in Paul's Slide is much larger than for the other landslides, the initial surface area of motion is comparable to the three other landslides. The signal detected by the network method arises from a combination of dynamic surface factors, not the eventual scale of failure. Since our analysis is based solely on surface movement, the volume ultimately displaced during a landslide is not an input for our calculations—underscoring that our method captures precursory surface behavior rather than relying on post-failure consequences.

This reinforces the value of network-based methods in revealing nuanced temporal patterns and transitions that would otherwise remain hidden when relying solely on conventional geophysical variables (Mei et al., 2025; Zhou et al., 2022; Wang et al., 2025).

## 5.3 Hydrologic Integration

To evaluate whether hydrological forcing could improve community detection outcomes, we incorporated precipitation data from PRISM (Oregon State University, 2015) in combination with velocity and slope (see Appendix A). However, no notable changes were observed in the community detection results. This lack of sensitivity may stem from the low resolution of PRISM grids, as well as limited data availability near the coasts, potentially rendering the changes too coarse for the community detection algorithm to detect.

To further test this hypothesis, we repeated the experiment using high-resolution precipitation input from a WRF-Hydro model (Li et al., 2023). Even with this improved spatial resolution, there was little improvement in the analysis. The strong correlation (0.67) between peak Z and precipitation data (see Fig. 3) suggests that community detection metrics already captures underlying hydrological mechanisms, or at least the surface hydrology, and therefore makes the inclusion of precipitation redundant. Advancements in in-situ soil moisture measurements could further improve the applicability of hydrological models, particularly for deep-seated landslide studies.

## 6 Conclusions

The extension of network science techniques from the Mud Creek case study to the broader Big Sur region yielded promising results. By subdividing the region into smaller sub-regions for varied terrain, we tested the effectiveness of this method as a

monitoring technique. The outcomes of community detection served as robust indicators of the landslides in W22-23, as seen in Fig. 2. Notably, the steady increase in  $Z_t$  leading up to failure, alongside the magnitude of  $Z_t$ , emerged as a crucial indicator. For instance, Paul's Slide exemplifies how outliers in  $Z_t$  can serve as early indicators of increased likelihood of landslides. Had this analysis been conducted in the relevant time frame, Paul's Slide, alongside Mill's Creek, Dani Creek Slide, and Gilbert's Slide, could have been identified as areas of concern, potentially allowing for preemptive monitoring and mitigation measures.

Our comparison of landslide susceptibility factors – landslide inventory, slope, cumulative displacement, precipitation, and InSAR coherence – with the outcomes of community detection, peak Z-score, underscores the importance of integrating multiple data sources. Each factor taken alone does not yield enough information to predict landslide, highlighting the need for comprehensive analyses. Furthermore, incorporating the entire temporal period captured by InSAR into the multilayer network improved classification between stable and hazard-prone sub-regions. However, it is crucial to ensure that remote sensing data accurately capture ground surface deformation for statistical analyses to be reliable.

Overall, the physically informed framework developed here—relying on measured creep velocity and slope—demonstrated strong potential for enhancing landslide hazard assessment. In particular, network metrics such as the peak Z-score offered a sensitive and scalable means of identifying emerging instability, even prior to failure. These findings support the utility of community detection techniques as a complement to conventional geophysical indicators, paving the way for improved, near real-time monitoring systems that can generate dynamic hazard maps and inform timely risk management strategies.

Code and data availability. The extents of the sub-regions are archived on DataDryad (Desai et al., 2024). Copernicus DEMs are available at https://spacedata.copernicus.eu/collections/copernicus-digital-elevation-model. On-Demand InSAR products were downloaded via Alaska Satellite Facility's HyP3 platform, available at https://hyp3-docs.asf.alaska.edu. Sentinel-1 data are available at https://search.asf.alaska.edu/, the ASF data search vertex. The full list of interferograms used are archived on DataDryad (Desai et al., 2024). The Miami INsar Time-Series software in PYthon (MintPy) is available at https://github.com/insarlab/MintPy. Precipitation data is provided by Parameter-elevation Regressions on Independent Slopes Model (PRISM) and is available at https://prism.oregonstate.edu/. Daily precipitation time series is taken for each of the sub-regions. The code used for creating the multilayer networks is available at https://github.com/vddesai-97/networkLandslide, and running the community detection algorithm is on https://github.com/GenLouvain/GenLouvain. The multilayer networks (nodes, edges, weights) for each of the sub-regions are archived on DataDryad (Desai et al., 2024).

### Appendix A: Inclusion of hydrological information into the multilayer network

Landslide studies often use precipitation data from (1) site-specific rain gauges with limited spatial coverage and nonuniform temporal resolution, or (2) spatially continuous interpolated gridded rain data. Effective and efficient monitoring of hydrological data has not yet caught up with remote sensing technology to produce high spatial and temporal datasets. Satellite-derived data, such as Soil Moisture Active Passive maps, which use passive microwave techniques and remotely sensed surface soil moisture on a global scale, underestimates in heavily vegetated areas (Das et al., 2019; Reichle et al., 2017; Fan et al., 2020). Remote sensing techniques only detect surface-level soil moisture, and process-based land surface models typically extend the

soil moisture estimates to one to two meters below the ground surface but have low spatial resolution (Koster et al., 2009). Therefore, we consider two hydrological datasets: PRISM (Parameter-elevation Relationships on Independent Slopes Model) for precipitation and WRF-Hydro (Weather Research and Forecasting Hydrological modeling framework) for soil moisture and precipitation on the Big Sur Coast.

320

**PRISM** The PRISM Climate Group develops spatial climate datasets using various monitoring networks and modeling techniques (Daly et al., 2008). These datasets include daily, monthly, and annual precipitation, and minimum and maximum temperatures for the contiguous United States. PRISM interpolates station measurements using a climate-elevation regression model that considers factors such as coastal distance, topography, and atmospheric conditions. There are about 13,000 stations that collect precipitation data and 10,000 for temperature. PRISM datasets have shown improved results for mountainous and coastal regions of the western United States, including our study sites.

*WRF-Hydro* WRF-Hydro is an open-source, physics-informed hydrological model (Gochis and Barlage, 2020). The model disaggregates precipitation at the land surface and simulates landslide-relevant processes such as water table depth, infiltration, subsurface lateral flow, and soil moisture using information like soil type, topography, and antecedent conditions. Li et al. (2023) utilized WRF-Hydro to simulate soil moisture within the Big Sur Coast region, incorporating seven in-situ soil moisture stations and nine USGS stream gages. This region has a complex terrain with heterogeneous vegetation, elevation, and slope. The data used in this study has a default soil column with a depth of 2 meters, divided into four layers: 0-10 cm, 10-40 cm, 40-100 cm, and 100-200 cm. Li et al. (2023) demonstrated that WRF-Hydro outperforms many established soil moisture products through data-informed methods that improve soil parameters.

The two datasets differ in resolution and the type of hydrological forcing they represent. PRISM has a  $4~\rm km^2$  resolution with daily precipitation outputs in mm, while WRF-Hydro has a  $1~\rm km^2$  resolution with outputs of soil moisture at different depths.

We considered soil moisture, water table depth, and precipitation as additional information to incorporate into the multilayer network as weights in addition to velocity and slope. There was insufficient difference in the community persistence signal when including hydrological information of any type. This is likely because velocity already incorporates underlying hydrological mechanics. When there is enough water in the soil, frictional resistance reduces, causing slow-moving hillslopes to speed up. As the soil dries, the hillslopes slow down. Since this information is already included in the multilayer network, adding hydrology data is redundant. Another reason the hydrology data might not be useful is that Mud Creek is a deep-seated landslide, and the data only went to 200 cm below the surface. To test the effects of adding in hydrological information on the community persistence of the 17 study sites, we applied precipitation data from PRISM (chosen for its success in the Western U.S.) as one of the weights, along with velocity and slope, for the multilayer network. Fig. A1 shows the mean community persistence  $\Pi$ , as discussed in the paper. The results for the multilayer network with weights w = vs, where v is velocity and sis slope, is shown in Fig. A1[a] and Fig. A1[b] shows the results for weights w = vsp, where p is precipitation from PRISM. We observe that including precipitation as a weight shows minimal differences in the community detection.

Figure A1. Geographical distribution of Z-scores. Mean Z-Score for [a] weights w = vs compared against [b] weights w = vsp. The white dots are the landslides.

325 *Author contributions.* VDD developed and carried out the analysis, and KED and ALH assisted throughout the development and analysis. ALH prepared the InSAR data. VDD prepared the paper and figures, and ALH and KED provided comments and edits.

Competing interests. The contact author has declared that none of the authors has any competing interests.

Acknowledgements. This work was supported by NSF grant PREEVENTS ICER-1854977 and we are grateful for conversations with our collaborators on this project. We also acknowledge the WRF-Hydro analysis conducted under this grant by Chuxuan Li and Daniel E. Horton, the results of which were used in Appendix A. Part of this research was carried out at the Jet Propulsion Laboratory, California Institute of Technology, under a contract with the National Aeronautics and Space Administration (80NM0018D0004), and supported by the Earth Surface and Interior program.

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
