# Peer review of "Using Network Science to Evaluate Landslide Hazards on Big Sur Coast, California, USA"

_EGUsphere, 2024_

## Author Comment (AC1)

**Reviewer #1 Comments**

We appreciate the reviewer's time and thoughtful feedback. We agree that the readability of the manuscript would benefit from modifications to whether particular items are described in the results, discussion, or conclusion sections. Additionally, we are happy to both expand on the discussion of network science itself, clarifying the unique value this method brings to the problem, and further elaborate on the monitoring implications of our analysis. These revisions will significantly enhance the clarity and overall impact of the work. Below, we respond to each of the comments and questions in detail.

Thank you for your innovative work. I have some comments and questions that I listed below:

Line 2 (and throughout the paper): the term "vulnerability" might be misused in the context of risk science (e.g. no estimate is made of infrastructure fragility). I would argue that the terms to be used are "hazard" and "exposure" depending on the context.

> We had introduced the term 'vulnerability' since we had created a new metric and it seemed appropriate to name it. Given the concerns of both reviewers, we will replace `vulnerability' with `hazard' as the context requires, based on the following definition:
>
>> "Landslide hazard maps indicate the possibility of landslides occurring throughout a given area. An ideal landslide hazard map shows not only the chances that a landslide might form at a particular place, but also the chance that it might travel downslope a given distance." - USGS (https://www.usgs.gov/faqs/what-a-landslide-hazard-map)

Line 13: Palmer (2017) refers to slow-landslide specifically and the definition of which specific process the current paper is addressing is not clear at this point

Line 22: please see my comment above - "mass movement of material (rock, earth, debris) down the hillslope, defined as a landslide event". It would be good to define specifically the process under study here. It is not only a semantic problem as rock fall, landslides, debris flow processes are mechanically and spatially different.

> Regarding both the Line 13 and Line 22 comments: we agree that further clarification of the specific process would be helpful to interpreting the results. Unfortunately, since there are no published studies of these recent landslide

events, except for Paul's Slide, we are not sure if all 4 of these events are associated with slow-moving landslides. To provide the full information available, we will note that all 4 of these landslides occur deeper than 1 m below the surface soil, and clarify that this study encompasses both slow-moving and debris slide processes.

Line 26-28: Can you check the https://blogbigsur (Drabinski and Bertola) reference- I couldn't access the reference mentioned. Any other scientific publication available?

We have confirmed that the website we list https://blogbigsur.wordpress.com/2023/03/31/update-37-with-repairs-underway-travel-opportunites-still-abound-on-the-big-sur-coast/, https://blogbigsur.wordpress.com/2023/03/16/update-31-assessments-on-highway-1-continue-at-areas-damaged-by-most-recent-storms/, and https://blogbigsur.wordpress.com/2023/03/07/highway-1-at-mill-creek-to-open-by-end-of-march-pauls-slide-still-set-for-long-term-closure/ is still live as of May, 2025. These are reports from the California Department of Transportation which provides details of the event. This referenced source provides valuable documentation when formal scientific studies are unavailable. Additionally, the California landslide database found on https://www.conservation.ca.gov/cgs/landslides have recorded these events and we will add that citation to lines 26-28.

Line 29 : More background information about network science would be good to have here - e.g. Nodes, Edges etc. Why is a network framework interesting in this case?

Network science is one of the largest developments in applied math and statistical physics during the past few decades, a field with its own conference (NetSci) which reports on its application to many fields including the granular/amorphous materials (Refs Bassett et al., 2011; Kivela et al., 2014; Mucha et al., 2010; Papadopoulos et al., 2016; Porter & Gleeson, 2016; Porter et al., 2019 in the paper); this large and active community of researchers motivated the present work.

As for a specific motivation for this project, the underlying landslide material is indeed made up of grains, and in our original and successful application (Desai et al 2023) we had been primarily attracted to the ability of network science to reduce a complex problem to a description in terms of spatiotemporal relationships. The network science approach is to provide an overview of the state of the system in terms of a set of nodes connected by edges, where each edge contains quantifiable data about the relationships between the nodes, information which (in the case of landslides) is available from remote sensing

data. In our revision to the paper we will emphasize this context (as was done in our earlier paper, but we hadn't originally repeated it here.)

Line 31 : "two categories: stable and vulnerable. A region is considered vulnerable if it is likely to experience a landslide event" - see my comment above - "susceptible" might be more appropriate?

Unfortunately, the word "susceptibility" already has a meaning for researchers studying both landslides and the statistical physics of phase transitions. Therefore we decided not to use that terminology (in fact, this is what led us to the term "vulnerable"). As stated in the earlier comment on the same topic, we will adjust our language usage to use the word "hazard".

Figure 1: a minimap of the location in the state / country would be nice for a geomorphologic context

During revisions, we will add that in as an inset to Fig.1 (a).

Line 64: it is unclear to which "mass downslope movement" you are referring to here. The link between slow moving landslide and evidence of associated mass movement is missing

Given the confusion of 'mass downslope movement' and slow-moving landslides in addition to the uncertainty of the processes of each of the 4 landslide events, we intend to clarify that 'mass downslope movement' encompasses slow-moving landslides (Paul's Slide) and debris or rock slides (potentially Mill's Creek).

Line 70: 1) 40x40 m2 should be 40m2 or 40x40m, I believe 2) The temporal resolution is not explicit

1) We mean (40 x 40) m^2 and will add these parentheses during revision.

2) The temporal resolution is irregular. Sometimes it is 12 days and sometimes 6 days, corresponding to Sentinel-1 passes, and we will add an explicit mention of this during revisions.

Line 90: How much of the total area represents the mask? Can it impact the analysis?

The mask represents about 1.7% of the total area. For any of the 17 sub-regions, the mask represents 4.8% on average, with the min being 0% (sub-region #1) and the max being 10% (sub-regions #7,8,10). We will add a summary of this

during revisions. Through tests done on an artificial system to mimic InSAR, we found that this does not impact the analysis since (1) it accounts for a small amount of the area and (2) the network analysis is robust to this choice of mask.

Line 105: A representation of the graph and communities would be important for the comprehension

We have a figure that represents the graph and communities in *Desai et al., 2023* (https://doi.org/10.1103/PhysRevE.108.014901) Fig 3. We will reference this specific figure in the paper.

Line 106: A Poisson sampling with Delaunay triangulation is unlikely to follow a hydro-geomorphologic logic - would there be an advantage in using slope units as the basis for the nodes/edges for example (https://doi.org/10.1016/j.geomorph.2020.107124)?

Thank you for sharing that paper, which is an interesting approach. Our choice of Poisson sampling with Delaunay triangulation achieves a similar purpose of encoding the underlying heterogeneity and objective delineation at a much higher spatial resolution (20 m) than slope units would. By factoring in slope steepness, we incorporate the topography within the weighted network. Therefore, we don't see an advantage, but it would be worth trying in future work.

Line 109: "we calculated the average velocity and slope of any two connected nodes and set that as the edge weight" - Are the community distribution sensitive to a metric different than the average (e.g. Maximum, Skewness, Kurtosis)?

The partition into communities is (by design) sensitive to the metric chosen to set the edge weight, which is why we used physics-based quantities to weight our edges. The average velocity and slope of any two connected nodes captures how the inverse-viscosity and gravitational load influence the system. Through tests done on an artificial system to mimic InSAR, we found that choosing the maximum or minimum would introduce a high-sensitivity to noise inherently presented in InSAR data; this reduced the effectiveness of the method. Additionally, choosing the skewness or kurtosis is more about looking at the variability in these measurements, which is doesn't tell you how much the system is currently flowing

Line 113: More details are needed about the GenLouvain algorithm

The generalized Louvain method (Blondel et al., 2008) is a standard network science technique, with over 16,000 citations. As such, it has become standard

not to describe it in detail but instead to refer to the features for which it has been chosen. In the revision, we will add that it was selected because it divides a network into communities by identifying where the edge weights are stronger within the community than one would expect at random, and this is the feature we are looking for within our data .

Line 125: In the case of a catastrophic failure (Millions m3), I would expect several communities involved to remain stable while bordering communities would see an increase in Z values - correct?

No, what we observe is that bordering communities see a decrease in Z-value between time slices while involved communities see an increase in Z. Communities involved with the moving hillslope represent areas that have increasing velocity. This translates to a higher than average weight within the network (weight is velocity times slope). Since communities are partitioned by the mean weight of the network, an area that is moving faster than average is identified as a community, and will continue to be identified as long as the area has a higher velocity. The Z-value for that community will increase then. Areas that are moving slower than the mean will not be identified as a community, and will therefore see a decrease in Z-value.

Is there any (albeit rare) scenarios where the Z-score could, for example, average out and be misleading?

That is an interesting question, but we struggle to come up with a scenario in which this would be possible for the following reasons. The Z-score is computed for the entire layer instead of for each community in the layer, and it only measures the positive persistence of any community. Since the Z-score does not account for communities decreasing in Z-value within a layer and we look at the overall Z-score change between layers, we are not sure if there is a scenario which could be misleading.

Line 138: Can you be more explicit of what a Z < 0 would actually mean?

$Z_t$ considers the change in Z between any two time slices. So if Z decreased in the following time slice, then $Z_t < 0$, but if Z increased, then $Z_t > 0$. When $Z_t$ is negative, there is very little persistence in communities in time. In this system, this corresponds to the dry season where there is very little forcing detecting in the hillslope. See our response for line 125 for a more in-depth explanation. We will add this description to the paper.

Line 140-145: It is not clear to me over which period the average community persistence is calculated; is it a rolling average?

No, it is not a rolling average, but an average taken over the entire time. We will clarify that the average community persistence is taken over the entire period of 2015-2022. This could, of course, also be done as a rolling average if the technique were deployed as a monitoring tool.

Line 148: "Here, darker sub-regions represent higher peak Z-scores. sub-regions with a relatively stable Z-score had peak Z < 2.5. Within the sub-regions that showed increasing Z, some sub-regions have peak Z < 3, and some have peak Z " - can you be more explicit about the 1 to 4 categories shown on Figure 2c?

The 1 to 4 values are not categories, but rather a binning of the calculated peak Z score for that sub-region. We will modify this line of text to clarify that these are numerical ranges of values, and will connect the peak Z values defined in Fig. 2c to the results more clearly.

Figure 2b: the asterisks are really small

We will increase the size of the asterisks.

Line 159: The Multivariate analysis should probably be introduced in the methodological section with the result explained in the Result section. A Discussion section could then be added before the conclusion

We believe that introducing the reasoning and method of the multivariate analysis in the methods section makes sense. We will move the first paragraph in section 4.1 to the methods section with some rewording.

4.1 Multivariate Analysis: Can you clean this paragraph, as there are several discrepancies and it makes it hard to follow: e.g. "Community persistence exhibits positive correlations with mean displacement (-0.53)", "Moreover, precipitation has a strong positive correlation by 0.63 with precipitation"

We will revise this paragraph to make it clearer. We mean to state the precipitation has a strong positive correlation of 0.63 with the landslide events.

Figure 3: Could you add a ROC plot and AUC score from the Z-score and events?

Below, we plotted the peak Z-score for each of the 17 subregions, calculated for the Nov 2015 to Dec 2022 dataset which is the focus of this paper. Red indicates a landslide event, and blue indicates a stable region. There is a sharp boundary

between high-Z (Z > 2.5) and low-Z (Z < 2.5) events, such that an ROC plot would simply be a step function. We will include this information in the revised manuscript or Supplemental Material, either as a plot like this or in some other format.

[Figure]

If we instead combine the analyses from both the Nov 2015 to Dec 2022 and Nov 2015 to Feb 2023 periods, as well as separately plotting the Mud Creek event from 2017, then a similar plot [see below] shows only one Z-score which differs from that classification. This also makes an ROC plot an inappropriate tool: for any reasonable choice of threshold near Z=2.5 there is just this one outlier.

[Figure]

The outlier is the Gilbert's Slide, which was identified as a hazard in the Dec 2022 period, but fell below the threshold a month before it catastrophically failed (mentioned on line 157). All other events fall above Z = 2.5.

Line 202: "we analyzed two time periods:Nov 2015 to Nov 2022 and Nov 2015 to Feb 2023" the two analysis periods remains unclear in term of their relationship or purpose

> Yes, the two time periods were chosen to show how analysis before and after the landslide events compare. We will add clarifying language to the paper to make that clear.

Line 206: The 97% is coming out of the blue and not convincing (as you pointed out) and consider, from what I understood, a single threshold. See above my comment on the ROC curve. Could you iterate the threshold with various scoring metrics to identify an optimal threshold and provide a better selling point for your method?

> The 97% comes from the calculation that during the two time periods we ran this analysis for: Nov 15 to Dec 22 and Nov 15 to Feb 23, there was only one time when the analysis mis-identified a region as stable: Gilbert's Slide. This region was correctly identified as a hazard in Dec 2022, but fell below the threshold in Feb 2023 right before it catastrophically failed in Mar 2023 (but only when using the longer time period)

> For both of the (overlapping) time periods, each of the sub-regions is either classified as stable or a hazard, depending on if a region experienced a landslide event in the months following the analysis. The 97% comes from 1 false negative out of 34 measurements (with a statistical complication that we are including each sub-region twice, once for each fresh analysis).

> The reason for the single threshold is described above, and there is not another choice that is reasonable to make and would adjust the observed percentage. Since there is only one subregion (out of 17) causing the false negative, and its Z-score is far below any reasonable choice of threshold, considering sensitivity and specificity would not improve our classification: this subregion is an outlier.

> We will make this calculation, and the lack of flexibility to choose a better threshold, clear during revisions.

Line 217: Correct "(Oregon State Univeristy, 2015) with velocity and slope in S2" - presumably Supplementary Information 2?

> Yes. We will correct that.

Line 221: What is the result of the WRF-Hydro model doing in the Conclusion section? it seems out of place

This placement arose because we observed a negative result, and this placement originally seemed like a way to mention it more briefly than we did for the positive results. Since both reviewers have suggested that it be included as a main result, we are happy to instead include it in the Results section.

The Conclusion looks more like a Discussion + Conclusion and the conclusion lacks specific recommendations for practical implementation. Overall the landslide inventories and their uses remains unclear and need to be addressed

In the revised manuscript, we will reorganize these sections to distinctly separate the discussion of the results from the conclusion. Additionally, we will expand the Discussion section to more thoroughly interpret the results and contextualize the methods used, especially incorporating the specific discussion points raised by both reviewers. We will place emphasis on how the network science techniques simplifies a complex system and adds to identifying the transition from stable to hazardous, as well as add in more about the potential applications of this method as a monitoring tool. We appreciate the thoroughness of your comments to clean up the language for better clarity throughout the paper.

---

## Author Comment (AC2)

**Reviewer #2 Comments**

We appreciate the reviewer's time and thoughtful feedback. We agree that the readability of the manuscript would benefit from modifications to whether particular items are described in the results, discussion, or conclusion sections. Additionally, we are happy to both expand on the discussion of network science itself, clarifying the unique value this method brings to the problem, and further elaborate on the monitoring implications of our analysis. These revisions will significantly enhance the clarity and overall impact of the work. Below, we respond to each of the comments and questions in detail.

This manuscript explores the use of network science to analyze landslide failure potential along the Big Sur Coast, with a focus on the concept of "community persistence" across subregions. While the use of network science is innovative and the figures are visually appealing, I have significant concerns regarding terminology, scale, and the clarity and coherence of the manuscript's structure.

**Major Comments**

The use of the term *vulnerability* is bizarre. In the risk assessment framework, *vulnerability* has a well-defined meaning —people, property, infrastructure, and resources, or environments that are particularly exposed to adverse impact from a hazard event. The authors appear to conflate this with *hazard*, which would be more appropriate terms in the context of slow-moving landslides. This issue persists throughout the manuscript and should be addressed comprehensively.

We had introduced the term 'vulnerability' since we had created a new metric and it seemed appropriate to name it. Given the concerns of both reviewers, we will replace `vulnerability' with `hazard' as the context requires, based on the following definition:

"Landslide hazard maps indicate the possibility of landslides occurring throughout a given area. An ideal landslide hazard map shows not only the chances that a landslide might form at a particular place, but also the chance that it might travel downslope a given distance." - USGS (https://www.usgs.gov/faqs/what-a-landslide-hazard-map)

The analysis is conducted at a subregional scale of 5 km², yet the landslide processes typically affect areas of a few 0.1 km². This mismatch raises questions about the sensitivity and appropriateness of the method for capturing relevant slope dynamics. Additionally, the definition of subregions is not clearly

justified—especially the inclusion of "varied terrain" within single units. From a monitoring perspective, this scale is difficult to reconcile with actionable insights, and no clear rationale is provided for not working at the scale of slope units or other, more geomorphologically meaningful divisions.

This should as well form a clear part of the manuscript discussion.

> We agree that this choice of length scale is important to establish on firm footing, and did some tests of subregional areas ranging from 25 km² to 5 km². The key argument for conducting our analysis at 5 km² instead of a couple of 0.1 km² lies in the possible application of this technique as a monitoring tool that is computationally efficient (a 250x larger set of grid points is a significant increase in computational time). We considered smaller subregion areas but this would have reduced the square footage of stable slopes -- an essential factor for identifying the slope dynamics that signal a transition to catastrophic failure.

> A potential future direction could be to expand this approach at a smaller spatial scale to determine if the method would improve in sensitivity and specificity. This could improve the applicability of this work for monitoring, at the cost of additional computing time. Were this technique to be under development as a monitoring tool, a careful analysis of the tradeoff between increased spatial resolution and increased computational time would be important to tackle before deployment.

From my perspective, this methodology making use of network science/community presence/nodes/etc. seems overly complex without delivering clear added value. Especially since, ultimately the Z-score based on the 'community persistence' metric only considers slope and surface velocity... Why not include the other factors introduced in the paper ? The exclusion is neither explained nor justified.

> In our original (and successful) application of this method (Desai et al 2023), we were drawn to the ability of network science in reducing a complex problem to a description in terms of spatiotemporal relationships derived from readily available remote sensing data. This approach allows us to study slow-moving landslides and identify the transitions between stable and unstable using minimal inputs.

> The value of simplifying the system to a network is demonstrated in the Multivariate Analysis (Section 4.1), which connects the results of community detection to known physical drivers of slope activity.

> Regarding your question on including other factors: the two inputs to the network -- surface velocity and slope -- were selected deliberately. The

inclusion of precipitation, which we also thought would be beneficial, is addressed in the Supplemental Material: we instead found that its inclusion did not significantly improve the community persistence metric. As such, we determined that it did not add value in this analysis. The other factors -- InSAR Coherence, Mean Displacement, Max Displacement -- were not used as inputs but rather served as benchmarks for evaluating the community detection results. Specifically, we used them to assess how well the Z-score aligned with these known geophysical measurements. Including these variables as dynamic inputs would introduce redundancy: coherence was already applied as a mask to the InSAR data, and velocity was one of the network inputs. Therefore, they were excluded from the network inputs to entangling cause and effect.

The goal of this method is to identify key transitions using the most reliable and widely available data. Our results suggest that network-based analysis using only slope and velocity is sufficient to capture the essential dynamics of landslides: its simplicity is one of its strengths.

We will revise the manuscript to better highlight the rationale for the use of network science, as we did in Desai et al., 2023, and also to better-highlight how we came to the conclusion that many factors could be excluded without significant loss of predictive power.

The manuscript would benefit from significant restructuring. Results and discussion are overly interwoven, and there is actually little discussion of the results/methods. Also, the Conclusion introduces new analyses rather than synthesizing findings. This undermines the clarity and scientific rigor of the narrative. Moreover, several key analytical steps—such as the multivariable analysis—are presented in the Results rather than the Methods section, reducing transparency.

We will separate the discussion from the conclusion, and the results of including precipitation with velocity and slope will be included in the results section instead. The multivariable analysis will be split to incorporate the first paragraph of 4.1 under the methods section instead and the rest will stay under results.

It must be made clearer from the abstract and throughout the manuscript that the methodology targets large, deep-seated, slow-moving landslides. The focus on four specific landslides (e.g., W22-3) is not well-motivated in the Introduction.

The focus of the 4 landslide events is that they were recent landslide events for which we had InSAR data for and could test the method on. We will add language to identify the 4 events as true positives. We will add in

more language to clarify this throughout and to highlight the motivation of studying these landslide events.

**Section-specific comments**

**Abstract**

What do you mean by 'which provides a natural way to incorporate spatiotemporal dynamics'?

We were trying to state that the network analysis provides an intuitive way of studying the spatiotemporal dynamics. We understand how that is not clear and will remove 'natural' from the sentence.

**Introduction**

The introduction could be more comprehensive, particularly in contextualizing the relevance of network science to landslide analysis.
Paragraph 30: Clarify what network science techniques entail and justify their application here.

We had aimed to keep this paper shorter, since the original paper (Desai et al 2023) had already covered the relevance of network science. Since both reviewers would like to have that information repeated in this paper, we are happy to add it to this paper as well. Among the key points are that network science has applications in many fields including granular/amorphous materials (Refs Bassett et al., 2011; Kivela et al., 2014; Mucha et al., 2010; Papadopoulos et al., 2016; Porter & Gleeson, 2016; Porter et al., 2019 in the paper).

The landslide material underlying a hillslope is made up of grains, and due to the success of applying network science in granular systems, we were attracted to the ability of network science to reduce this complex system to a description in terms of spatiotemporal relationship in our original and successful application (Desai et al 2023). This is done by providing an overview of the state of the system in terms of a set of nodes connected by edges, where an edge contains quantifiable data about the relationships between the nodes, information which (in the case of landslides) is available from remote sensing data. In our revision to the paper we will emphasize this context (as was done in our earlier paper, but we hadn't originally repeated it here.)

Again, reconsider the use of *vulnerability*.
Instead of the term vulnerability, we will use hazard, as defined by USGS.

Clarify the rationale behind focusing on the W22-3 landslides.
It is not clear in the introduction why you focus on the 4 landslides of W22-3 and not the 44 others.

We look at all 44 landslides, but only 4 of them failed catastrophically and as such are true positives. Therefore, we will reword within the paper to 4 true positives and 40 true negatives in this study for the W22-23 period.

**Data**

Paragraph 50: The initial detection of 44 active landslides is mentioned twice and then largely ignored. Either omit or incorporate this information meaningfully in the analysis—e.g., assessing their impact on community persistence in other subregions.

As described above (comments on Introduction), we will be making a change to include both populations explicitly.

Paragraph 60–65: Rather than stating volumes removed, provide basic information about landslide size. Paul's Slide, for example, appears much larger than others, which likely impacted the analysis. This should be discussed in the discussion section as well.

Due to a lack of clear classification by landslide type, we decided to focus on providing information which was available for all four landslides, and one of the available metrics was the volume removed. There are no published studies of these recent landslide events, except for Paul's Slide, and we are not sure if all 4 of these events are associated with slow-moving landslides. We will define that these landslides occur deeper than 1 m below the surface soil. We will add other references and clarify that this study encompasses both slow-moving and debris slide processes.

We agree that Paul's Slide is much larger than the other in terms of volume removed. The initial surface area is roughly similar to those of the other three landslides that failed. Since only the moving surface of a hillslope is being detected in the network analysis, the volume removed has no influence on the results. This is a good discussion point and we will add the above explanation to the manuscript.

It feels premature to present the number of landslides detected before explaining the InSAR methodology.

We agree that moving Sec 2.2 before Sec 2.4 could be helpful and will implement this change when writing the revision.

Figure 1: Increase the size of panel (a), reduce legend clutter, and adjust scale values in panel (c) for cleaner presentation.

We will take this into consideration when we modify this image to improve its presentation based on both reviewer's comments.

The term "InSAR snapshots" is unclear—consider replacing them with "deformation maps" or similar.

Our goal is to keep terminology as easy for the reader as possible and agree that deformation maps (or something akin) could help achieve this.

How did you defined you subregions, is it relevant to include 'varied terrain' in a subregion? Also, in a monitoring perspective, how to deal with information at a 5km² scale?

We defined the subregions as areas of (5 x 5) km$^2$ that were along the coast of the defined area, in which landslides had been identified by co-Author Alexander Handwerger and InSAR data had been processed.

It is relevant to include regions of varied terrain as it is important for community detection to have areas of stable hillslopes to compare to hillslopes within the same subregion that are moving and transitioning from stable to unstable. They are the control that we measure against.

By quantifying the results of each subregion, this method has the ability to be used in a monitoring perspective by identifying areas that have a high likelihood of experiencing a landslide..

Paragraph 115: Why are only slope and velocity used in community persistence? This choice should be justified more explicitly.

The inclusion precipitation does not enhance or improve the results, an observation that is discussed in the both paper and Supplemental Materials. See responses above under 'Major Comments' for further details about how we will clarify and emphasize this during the revisions.

Paragraph 130: "We use peak Z to quantify differences between sub-regions to better classify the slow-moving landslides as stable (peak Z < 2.5) or vulnerable (peak Z > 2.5)." Clarify whether you are classifying subregions or landslides based on peak Z.

In paragraph 130, we will clarify the sentence to state that while peak Z is used to classify sub-regions, it is an indicator of the landslides within that sub-region.

Landslide susceptibility is approximated using the sole slope. Why not using susceptibility models – I guess there are many available for the region?

No, while there are many susceptibility maps available in this region, susceptibility maps do not quantify the probability of a near-real-time

landslide occurrence for a given area. Through the inclusion of velocity along with slope, we provide near real time monitoring of landslides in the region.

**Results**

Figure 2: Clearly show the correspondence between Zt point size and value. Make the color scheme and value thresholds more intuitive (e.g., emphasize the vulnerable vs. stable divide).

We will consider this and see if we can make the interpretation more intuitive while still keeping the colors accessible in monochrome and for folks who are colorblind (this colormap was chosen to achieve both of these aims).

How do you rule out that Paul's Slide is detected so clearly simply because it is much larger than the others? This point deserves to be discussed.

While the volume removed in Paul's Slide is much larger than in the others, the initial surface area is roughly similar to those of the other three landslides that failed: the signal we are detecting comes from the same combination of factors collectively to identify it as a hazard. Since only the moving surface of a hillslope is being detected in the network analysis, the (eventually-known) volume removed has no influence on the calculation we do. This is a good discussion point and we will add the above explanation to the manuscript.

**Discussion & Conclusion**

Results, discussion and conclusion are overly interwoven, and there is ultimately little discussion of the results/methods.

We agree that the results, discussion, and conclusions sections could benefit from clearer separation. In the revised manuscript, we will reorganize these sections to distinctly separate the discussion of the results from the conclusion. Additionally, we will expand the Discussion section to more thoroughly interpret the results and contextualize the methods used, especially incorporating the specific discussion points raised by both reviewers. We will place emphasis on how the network science techniques simplifies the system and the value it adds to identifying the transition from stable to unstable.

How did you define the subregions? Is it relevant to include 'varied terrain' in a subregion?

See response above under 'Major Comments'.

Also, from a monitoring perspective, how to deal with information at a 5km² scale (e.g., to 'potentially allowing for preemptive monitoring and mitigation measures')?

See responses above under 'Data'.

How does changes in slope velocity (of a few cm/yr) over a few 0.1 km² influence velocities max/averaged over subregions of that size? Why not working e.g., over slope units?

The average velocity is taken for any two nodes connected by an edge, which spans roughly 20 m, not for the entire subregion. Each edge has a slope value which accounts for the slope at a higher resolution at roughly 20m than a slope unit would.

paragraph 205: The Conclusion should not introduce new results; instead, it should synthesize key findings and implications.

We will move the results of the precipitation inclusion into the results section.

**Final Recommendation**

This manuscript addresses an important (and complex) aspect of landslide risk, aiming at combining satellite remote sensing and network science framework to provide a comprehensive monitoring technique. However, I see fundamental issues with terminology, scale, methodological clarity, and manuscript structure that significantly limit its current impact.

Thank you for your time and thoughtful review. We appreciate the feedback and believe this paper will improve on its clarity and impact once we implement these changes to the structure, vocabulary, and expansion on the decisions we made above.

---

## Author Response (AR2)

**Reviewer #1**

This is quite a thorough revision of the manuscript, and it's clear the authors have made a commendable effort to address many of both reviewers' previous comments — which is much appreciated. The use of more appropriate terminology, a clearer structure, and improved explanation of network science concepts have all significantly strengthened the manuscript.

That said, I have a few remaining suggestions. As a general comment, I would recommend, in the discussion section, to link the findings more explicitly to relevant literature to better position the study within the broader context and help readers understand how it advances current knowledge.

Thank you for your review. We appreciate the recommendation to link our discussion to relevant literature as it would greatly improve the context of the study. We have linked the findings to relevant literature in the most recent manuscript in the Discussion section, as well as added more citations in our introduction to link our study to recent research. Below, we have addressed the more specific comments, and believe that this manuscript has a clearer flow and improved impact.

Below are some more specific, paragraph-by-paragraph comments:

• L25–30: This paragraph would benefit from further rephrasing. For example, simply stating that these are deep-seated landslides would suffice. Also, four deep-seated landslides were recorded by USGS, but others may exist. I don't think it is necessary to mention that the specific landslide process has not been studied. Also, the term "shallow acceleration" is unclear — please clarify or rephrase.

We have rephrased this paragraph to simply state the landslides are deep-seated. We replaced shallow acceleration to "reactivated where some material from the surface failed and engulfed the highway".

• L35–40: As pointed by the question from the other reviewer, network science is not a familiar concept for many geomorphologists, and further details in the intro are welcome. Yet I believe it could be made more useful – not by expanding it but by improving how the concept is introduced. Briefly explain what network science is, why it is potentially useful in this context, and how it adds value compared to more traditional approaches (e.g., early warning systems, other monitoring techniques). Also, I remain skeptical about the connection drawn between the granular nature of hillslopes and the predictive power of network science in granular materials. The method's strength seems to lie more in its ability to simplify systems by reducing the number of required parameters, rather than in any link to grain-scale processes.

We have added additional details in L35-40 to improve on the introduction of network science. The granular nature of hillslopes and the studies reference show that granular failure does not occur spontaneously, but that there is a precursory signal for the transition from stable to unstable. This signal can be identified using network science. We have rephrased some of the language as well as added a sentence in L35-45 to clarify this importance.

• L55: Consider revising "satellite coherence" to "InSAR temporal coherence," as the former is typical for SAR data.

We replaced "satellite coherence" to "Insar temporal coherence" to be consistent.

• L60: You mentioned in your rebuttal that 'coherence was already applied as a mask to the InSAR data', so can you write here that you 'include InSAR temporal coherence as a variable to measure satellite reliability'?

Line 60 already states this verbatim.

• L90: The Copernicus DEM has a native resolution of 30 x 30 m. If you downsampled it to match the multi-looked Sentinel-1 data, it's better to state that explicitly (avoid phrasing like "provides elevation data at 40 x 40 m," which could be misleading).

We corrected this part of the sentence to be "downsampled to \$40 \times 40\$ \unit{m}\$^2\$ resolution (matching to InSAR resolution) \citep{hyp3}".

• L120: I'm still unclear about the rationale for using the volume of material removed during reconstruction works, instead of (even rough) estimates of landslide size volume — which would presumably be a more direct and consistent proxy. The quantity of removed material likely varies based on many context-specific factors (e.g., road design, access needs), making it a less reliable comparative metric.

We agree that estimates of landslide size volume would be a more direct and consistent proxy, but that information is not available. Therefore, using Google Earth, we measured the surface area of the landslide and have added that information on L120 instead. We have removed the volume of material removed to keep the text clear and focused.

• L135: This point may not have been clearly communicated in my previous comment, so I'll rephrase: why use slope as a proxy for landslide susceptibility when actual landslide susceptibility maps are available? While this may not significantly alter your results, using such maps could provide a more comprehensive view, as they integrate multiple contributing factors beyond just slope.

Landslide susceptibility maps can provide a more comprehensive view than slope maps alone, but looking at the USGS susceptibility map, we observe that

most of the area has high susceptibility, which would provide many false positives. By using a slope map, we have a data-driven approach that utilizes gravitational driven stress by integrating how steep a slope is, distinguishing between hillslopes, and therefore allowing the algorithm to identify patches of area that have a higher chance of experiencing a landslide due to recent climatic conditions rather than identifying whole acres of area. In other words, a slope map will output a spatial location on the scale of a landslide.

**Reviewer #2**

The research proposed by Desai and co-authors is an interesting contribution to the study of (slow moving) landslide hazard with network science. I was not involved in the previous round of review. However, for assessing the current version of the manuscript, I have looked at the previous comments and the revised version.

In their responses to the other two reviewers, the authors have made an important revision of their work, addressing a large part of the concerns/suggestions of the first round. In its current version, the manuscript is to me almost ready for publication, pending a few "minor" points that I have listed below:

• In the introduction, I would welcome a few extra words on network science (what is it, why using it here in the specific case of landslide hazard?) since most readers may not be familiar with this concept.

We have added additional details in L35-40 to improve on the introduction of network science.

• Lines 25-30 and lines 120\_130, and also elsewhere in the manuscript, the term landslide events is used. In the literature, one commonly have in mind that landslide events are made of several landslide features. However, here; for the landslide event of Paul's slide, it seems that there is only one landslide. And in figure2c, there are 4 landslides (1 per event) that are pictured. Somehow, that could be said/formulated in a clearer manner and say in a more explicit way how many landslide are in the event inventories and how many have been studied.

We have removed the term "landslide events" and instead cleanly stated that there are 4 landslides that are studied throughout the paper.

• Line 134. Slope is used as landslide susceptibility. Why not using a landslide susceptibility map instead? Maybe this methodological choice could be better justified.

Landslide susceptibility maps can provide a more comprehensive view than slope maps alone, but looking at the USGS susceptibility map, we observe that most of the area has high susceptibility, which would provide many false positives. By using a slope map, we have a data-driven approach that utilizes gravitational driven stress by integrating how steep a slope is, distinguishing between hillslopes, and therefore allowing the algorithm to identify patches of area that have a higher chance of experiencing a landslide due to recent climatic conditions rather than identifying whole acres of area. In other words, a slope map will output a spatial location on the scale of a landslide.

• One aspect that can still be improved is the discussion section where, to my opinion, reference to the broader literature/context is somehow missed.

We appreciate the recommendation to link our discussion to relevant literature as it would greatly improve the context of the study. We have linked the findings to relevant literature in the most recent manuscript in the Discussion section, as well as added more citations in our introduction to link our study to recent research.

Overall, well done to the co-authors for this interesting research on landslide hazard.

Thank you for your time and review. We have addressed your comments above and believe that this manuscript is much better for it.